# A Global Women’s Rugby Union Web-Based Survey

**DOI:** 10.3390/ijerph20085475

**Published:** 2023-04-12

**Authors:** Natalie Brown, Geneviève K. R. Williams, Anna Stodter, Melitta A. McNarry, Olga Roldan-Reoyo, Kelly A. Mackintosh, Isabel S. Moore, Elisabeth M. P. Williams

**Affiliations:** 1Applied, Sports, Technology, Exercise and Medicine (A-STEM) Research Centre, Swansea University, Swansea SA1 8EN, UK; 2Welsh Institute of Performance Science, Swansea SA1 8EN, UK; 3Department of Public Health & Sport Sciences, Faculty of Health & Life Sciences, University of Exeter, Exeter EX4 4PY, UK; 4Research Centre for Sport Coaching, Carnegie School of Sport, Leeds Beckett University, Leeds LS1 3HE, UK; 5Cardiff School of Sport & Health Sciences, Cardiff Metropolitan University, Cardiff CF5 2YB, UK

**Keywords:** women’s rugby, female, concussion, injury, questionnaire, coaches

## Abstract

Rugby Union (rugby) is a full-contact team sport characterised by frequent collision events. Over one third (2.7 million) of global rugby participants are women and girls. Yet, most rugby research, laws, and regulations are derived from the men’s game with limited transferability to the women’s game. This includes research focused on injury and concussion management. Greater insights are urgently required to enable appropriate adaptations and support for all rugby participants. Therefore, this paper presents the protocol for a project that sought to gather insights into the understanding, experiences, and attitudes of players and coaches in women’s rugby regarding key issues of concussion, injury, and training for injury prevention, as well as the implications of the menstrual cycle for training and performance. From August 2020 to November 2020, online, open, cross-sectional surveys for players and coaches were distributed globally through rugby governing bodies and women’s rugby social media platforms using snowball sampling. Survey responses were recorded anonymously via a GDPR-compliant online survey platform, JISC (jisc.ac.uk, Bristol, England). Participant eligibility included being ≥18 years and either actively playing or coaching women’s rugby 15s and/or sevens, or having done so in the past decade, at any level, in any country. To enhance the number and accuracy of responses, the survey was professionally translated into eight additional languages. A total of 1596 participants from 62 countries (27 ± 6 years; 7.5 ± 5.1 years of playing experience) and 296 participants from 37 countries (mean age = 36.64, SD = 9.09, mean experience = 6.53 years, SD = 3.31) completed the players’ and coaches’ surveys, respectively. Understanding women’s participation in and experiences of rugby is important to enable lifelong engagement and enjoyment of the sport and health during and following participation.

## 1. Introduction

Rugby Union (rugby) is a fast-paced collision sport, where the high incidence of brain and bodily injuries and their potential long-term consequences have been a key concern for many years [1,2]. Despite this, rugby has a global playing population approaching ten million, of which almost a third are women and girls [3]. The women’s game is the fastest growing area of rugby globally, with participation rising by 28% per year [3]. Regardless of the sport’s relative female representation, existing studies relating to physical and technical demands, preparation strategies [4], injury mechanisms, injury management, training, and return to play protocols are almost exclusively based on male players [5,6]. This reliance on male-derived data is consistent throughout medical and sport and exercise sciences research, with androcentric data often generalised to females [5] despite a dearth of evidence supporting the efficacy of this generalisation. In contrast, there is a growing body of evidence indicating that sex and gender differences may make such generalisations inappropriate and even unsafe [7,8,9,10].

In women’s rugby, concussions are one of the most common injuries, carrying a significant injury burden [11,12]. There is evidence that female athletes are 2.6 times more likely than males to suffer a concussion in sport [13,14,15], with a more severe and prolonged symptom burden [16,17]. A variety of intrinsic and extrinsic factors have been suggested as contributors to this greater injury burden, including better reporting of symptoms in females and sex differences in head, neck, and spinal anatomy and biomechanics [18,19] and neuronal structure [20]. Furthermore, for those with female reproductive organs, hormonal fluctuations throughout the menstrual cycle can affect injury and recovery [21]. Moreover, an increase in oestrogen and progesterone at the time of concussion has been associated with poorer post-concussion outcomes in females [21].

In addition to biological sex-specific contributors, there are multifaceted gendered, environmental, and contextual factors that influence player experiences of injury and recovery [22]. These include social norms, biases, and inequalities apparent in access to and engagement in training, competition, and treatment environments. For example, fewer opportunities to participate in field-based and collision sports may result in a relatively low ‘training age’ in female players, who may also have minimal access to facilities and expert support [23]. Very little is known about the knowledge, experience, qualifications, and injury prevention practices of the women’s rugby coaching workforce. 

In this paper, the term sex is used to refer to biological attributes associated with physical and physiological characteristics, whereas gender is used to refer to socially constructed roles, behaviours, and identities [24]. Given the differences in experiences, severity, and implications of injury according to player sex and gender interactions, there is an urgent need to create female- and women-specific evidence bases to inform the development of future strategies to support player safety and performance. Therefore, this paper presents the protocol for a project that developed a global survey to gain insights into players’ and coaches’ understanding and experiences in women’s rugby, to establish an evidence base regarding key issues of injury and concussion, which will be presented in future research.

## 2. Materials and Methods

The method is reported according to the Checklist for Reporting Results of Internet E-surveys (CHERRIES) checklist [25].

### 2.1. Research Design

Separate women’s rugby surveys were developed for players and coaches respectively, using an open, voluntary, cross-sectional design, intended for global participant cohorts. Survey responses were recorded anonymously via the General Data Protection Regulation (GDPR)-compliant JISC online survey platform (jisc.ac.uk, Bristol, UK). To enhance the number and accuracy of responses, both surveys were professionally translated from English into eight additional languages: French, Spanish, German, Italian, Japanese, Welsh, Cantonese, and Russian. Full coach and player surveys are accessible in English in Appendix A, respectively. The surveys were launched in August 2020 and remained open for 12 weeks, until November 2020. Despite data being collected during COVID, questions were worded accordingly to state responses to refer to ‘during your normal activities’. Both snowball and purposeful sampling methods were utilised [26]. Specifically, a promotional e-poster, with live links to both surveys in the nine languages, was shared on women’s rugby social media platforms and news websites. Contact with potential participants was also made via mailing lists from world and national rugby governing bodies.

Ethics approval was obtained from Swansea University Research Ethics Committee (reference number 2020–035), and the study was conducted in accord with the Declaration of Helsinki 2013. A participant information sheet was presented on the first page of the survey, followed by a consent form, which participants were required to accept to open the survey. Due to the anonymous nature of the survey, all data were confidential; participants were informed that they would not be able to withdraw their responses once submitted as a result. Moreover, due to the anonymity of the survey, specific identification of and intervention for participants with medical risk factors were not feasible.

For both the players’ and coaches’ surveys, inclusion criteria were being ≥18 years and regular players or coaches, respectively, of women’s rugby (rugby 15s and sevens) at any level, for any rugby club or organisation, in any country, in the past ten years. For the coaches’ survey, those with coaching or managerial roles, including strength and conditioning programming and involvement in the scheduling of training and match fixtures, were invited to participate.

Voluntary participation in this study was not incentivised. Participants were informed that by completing the survey, they would be making an invaluable contribution to the knowledge base of women’s rugby. Specifically, the information would be used with the aim of improving training strategies, practices, and guidelines specifically for women athletes. Items were presented to each participant in the same order; as this was an individual survey, completed independently, the order that questions were presented was not considered to introduce any potential bias.

### 2.2. Survey Format

The player survey included a maximum of 149 multiple-choice and short-answer questions, presented in three sections. Section One addressed demographics, playing positions, experience and level, and basic health information, with a minimum of 19 and maximum of 43 questions to answer, depending on former answers. Section Two included health-monitoring practices, preparedness for rugby collisions, concussion history, strength and conditioning, injury training prevention, and injury management, which were addressed through a minimum of 57 and maximum of 72 questions. Section Three, which included a minimum of three and maximum of 34 questions, focused on concussion symptoms specific to women, the effects of the menstrual cycle on training, performances and injuries, female physiology, medical support, and financial resources in women’s rugby. Prior to answering any questions in Section Three, players were asked what sex they were assigned at birth with subsequent questions tailored appropriately. Completion of the players’ survey was estimated to take up to 60 min, depending on the detail provided in the free-text questions. Logic was applied to the survey to ensure that only relevant questions were completed.

The coach survey also included three sections. Section One comprised 40 main questions about demographics, coaching and playing experience, coaching qualifications and level, knowledge and experience of, and attitude towards, concussion, and knowledge of training interventions, such as neck strength. Section Two comprised 11 questions focused on strength and conditioning practices and experience, coaching practice, including injury prevention training, and management strategies. Section Three asked about experience, knowledge, understanding, and perceptions of injury prevalence, and performance in relation to the menstrual cycle. Free-text questions sought any further thoughts and ideas around training strategies and resourcing required to advance women’s rugby in a safe and sustainable manner.

All questions were developed specifically to address the objectives of this study. Due to the breadth of these objectives, this precluded the use of previously validated questionnaires to avoid excessive participant burden and thus engender high attrition rates.

## 3. Results

### 3.1. Survey Respondents

A total of 1596 participants from 62 countries completed the player’s survey (age: 27 ± 6 years; playing experience: 7.5 ± 5.1 years; Table 1), and 296 participants from 37 countries completed the coaches’ survey (age: 36.6 ± 9.1 years; experience: 6.5 years ± 3.3) (Table 2).

Respondent IP addresses were not checked for multiple submissions from each participant as the survey team was contacted by individuals in some localities advising that not all players owned or had access to a connected device to complete the survey. In these situations, the same computer was used by multiple participants. Instead, the data were screened to check for duplicate entries from the same user based on country, age, height, body mass, age started playing, and years playing rugby. If any duplicates were identified based on these values, further answers were compared manually. No duplicates were identified.

Whilst some questions required an answer, such as the participant’s country, age, and anthropometric data, the majority of questions were optional and only visible dependent on pre-requisite questions. Participants were able to review and change their answers prior to submitting the questionnaire, but due to the anonymous nature, information provided could not be amended following submission.

Prior to the release of the survey, pilot testing was completed by players from a university women’s team and several players on their country’s national team. The survey length and scope were also reviewed by representatives from the international rugby governing body, and minor revisions were completed based on feedback from these representatives. Due to changes, the pilot data were not included in the final data set.

### 3.2. Response Rates

It is unknown how many individuals received or opened the survey.

### 3.3. Data Analysis

All questionnaire responses not completed in English were translated into English by a native speaker in each of the other languages and verified by a second native language-speaking individual for data processing and analysis. Descriptive and statistical data were analysed using IBM SPSS Statistics 38, with significance set at *p* < 0.05. Free-text responses were analysed using a qualitative description (content analyses), consisting of organising the text into content categories. Counting the frequency of words in text was completed for some free-text responses, and descriptive codes were assigned to data to identify raw data themes, guided by research questions. Next, these descriptive codes were considered together and grouped into more abstract categories by generating interpretative codes. Finally, relationships between interpretative codes were identified to help organise the categories. At each stage, the content analysis was shared and questioned by another member of the research team.

## 4. Discussion

This brief report describes the protocol used for a global women’s rugby survey. The survey aimed to collect demographic, training, and injury data, along with coach and player experiences and an understanding of issues central to the women’s game. The wealth of information collected in the survey, to be presented in a series of work, will facilitate sex- and gender-specific, evidence-informed actionable insights about training, injuries, and experiences based on player and coach perspectives. This knowledge can inform future support and practices required to improve the health and performance of women players. An online, open-access survey enabled global responses across a large dataset. This enabled a more representative sample of players and coaches across different levels of competition, in comparison to interviews or focus groups, which would allow for in-depth insights into particular settings or sub-samples of interest.

Based on a recent systematic scoping review of scientific evidence in women’s rugby [23], an expert-established consensus identified injury as one of three high-priority research themes. Specifically, concerns regarding concussion occurrence, risk factors, mechanisms, and return-to-play management and female-specific responses to concussion were reported [23]. Female health was also reported as a research priority, particularly relating to the menstrual cycle and physical performance [23]. In underpinning this future research, the current survey-based method will provide invaluable knowledge of the understanding, experiences, and attitudes of players and coaches in women’s rugby regarding the key issues of concussion, injury, and training for injury prevention, as well as the implications of the menstrual cycle for training and performance.

A key strength of the current protocol was that participants could choose from nine different languages in which to complete the survey. This increased the likelihood of participants being able to complete it in their first language and widened participant reach globally. Nonetheless, it is pertinent to note key limitations of this study protocol, including cultural nuances in the interpretation of and responses to questions. For example, understanding and comfort around sharing sensitive information relating to the menstrual cycle may vary according to culture. Indeed, as questions were not mandatory to complete, the data collected may be skewed by participants’ comfort in talking about the subject. However, this approach did allow all participants to take part in the survey and have their views represented on subjects with which they felt comfortable, rather than participants feeling unable to contribute to this research due to sensitivities around a topic area. The anonymous survey was designed to minimise such limitations.

## 5. Conclusions

This manuscript presents the protocol for a global survey, which was designed to understand player and coach experiences regarding key issues in women’s rugby. This protocol facilitates future research to establish an evidence base for training practices, the management of menstrual cycle issues, and injury prevention tools.

## Figures and Tables

**Table 1 ijerph-20-05475-t001:** Summary of the highest reported playing level and geographic region for participants in the player’s survey (*n* = 1594) and the percentage of participants for each level by region (by row).

Region	Total	Nat. Team	Prem. Club	1st Div. Club	1st Div. Uni.	2nd Div. Club	2nd Div. Uni.	Rec.
Europe	534	93 (17%)	54 (10%)	171 (32%)	15 (3%)	100 (19%)	25 (5%)	76 (14%)
United Kingdom	424	32 (8%)	40 (9%)	141 (33%)	44 (10%)	80 (19%)	17 (4%)	70 (17%)
Asia	98	40 (41%)	8 (8%)	23 (23%)	5 (5%)	5 (5%)	1 (1%)	16 (16%)
South America	67	33 (49%)	0 (0%)	19 (28%)	4 (6%)	7 (10%)	0 (0%)	4 (6%)
North America	249	38 (15%)	41 (16%)	68 (27%)	27 (11%)	45 (18%)	14 (6%)	16 (6%)
Africa	30	21 (70%)	1 (3%)	2 (7%)	3 (10%)	2 (7%)	1 (3%)	0 (0%)
Middle East	9	6 (67%)	0 (0%)	3 (33%)	0 (0%)	0 (0%)	0 (0%)	0 (0%)
Oceania	165	45 (27%)	50 (30%)	40 (24%)	3 (2%)	14 (8%)	1 (1%)	12 (7%)
Not stated	18	9 (50%)	1 (6%)	1 (6%)	0 (0%)	3 (17%)	1 (6%)	3 (17%)
Total	1594	317	195	468	101	256	60	197

Nat. Team, national team; Prem. Club, premier club; 1st Div. Club, first division club; 1st Div. Uni., first division university; 2nd Div. Club, second division club; 2nd Div. Uni., second division university; and Rec., recreational.

**Table 2 ijerph-20-05475-t002:** Summary of the highest coaching level and geographic region for participants of the coach survey (*n* = 296) and the percentage of participants coaching at each level from each region (by row).

Region	Total	Youth	Nat. Team	Prem. Club	1st Div. Club	1st Div. Uni.	2nd Div. Club	2nd Div. Uni.	Rec.
Europe	93	3 (3%)	7 (8%)	37 (40%)	26 (28%)	6 (6%)	10 (11%)	0 (0%)	4 (4%)
United Kingdom	59	0 (0%)	12 (20%)	18 (31%)	5 (8%)	7 (12%)	7 (12%)	0 (0%)	10 (17%)
Asia	23	0 (0%)	11 (48%)	8 (35%)	0 (0%)	1 (4%)	0 (0%)	0 (0%)	3 (13%)
South America	28	14 (50%)	2 (7%)	2 (7%)	6 (21%)	4 (14%)	0 (0%)	3 (0%)	0 (0%)
North America	49	1 (2%)	9 (18%)	26 (53%)	1 (2%)	8 (16%)	1 (2%)	0 (0%)	3 (6%)
Africa	2	0 (0%)	1 (50%)	1 (50%)	0 (0%)	0 (0%)	0 (0%)	0 (0%)	0 (0%)
Middle East	2	0 (0%)	0 (0%)	2 (100%)	0 (0%)	0 (0%)	0 (0%)	0 (0%)	0 (0%)
Oceania	35	0 (0%)	7 (20%)	19 (54%)	6 (17%)	1 (3%)	2 (6%)	0 (0%)	0 (0%)
Not stated	5	1 (20%)	3 (60%)	1 (20%)	0 (0%)	0 (0%)	0 (0%)	0 (0%)	0 (0%)
Total	296	19	52	114	44	27	20	0	20

Nat. Team, national team; Prem. Club, premier club; 1st Div. Club, first division club; 1st Div. Uni., first division university; 2nd Div. Club, second division club; 2nd Div. Uni., second division university; and Rec., recreational.

## Data Availability

The data presented in this study are available on request from the corresponding author; no data containing identifiable information will be shared.

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
