# Peer review of "A Global Women’s Rugby Union Web-Based Survey"

_ijerph, 2023, doi:10.3390/ijerph20085475_

Round 1
Author Response
Please see the attachment.
Thank you for the opportunity to amend our manuscript considering your comments. We feel that the readability and strength of our manuscript has improved and would like to thank the reviewers for taking the time to appraise our submission. We have provided a point-by-point response to each comment in the attached document. Response to Reviewer 1 comments are labelled comment 2.1-2.9.

Reviewer 2 Report
It is weird that this is labeled as a study protocol and yet is providing results of the study
Introduction
No comments
Methods
Please provide either the full surveys as online supplements or links in the body text to the surveys for other researchers to replicate. After reading further, if you do want to try to stick with the idea that this is a study protocol, the surveys absolutely must be included.
line 179: What do you mean by "critical friend?"
Since you have clearly finished the research, why aren't you presenting the full results here? One major reason to publish a study protocol is to get feedback on methods before conducting the research--it seems like this manuscript has little benefit anymore
No comments on the results or discussion sections as is, except what I just said above--since you've finished the study, just go ahead and present the results
Author Response
Please see the attachment.
Thank you for the opportunity to amend our manuscript considering your comments. We feel that the readability and strength of our manuscript has improved and would like to thank the reviewers for taking the time to appraise our submission. We have provided a point-by-point response to each comment in the attached document. Response to Reviewer 2 comments are labelled comment 3.1-3.5.

Round 2
Reviewer 2 Report
Thank you for your attention to comments and thorough responses. Below, please find my last minor comments and suggestions on the current draft:
Title still says "study protocol"
Line 40: like in your abstract, you should initially state "rugby union" and then can abbreviate to rugby
Discussion is much improved
Conclusion is acceptable
supplementary files were not viewable or hosted on the MDPI review page for separate download
Author Response
Thank you for taking the time to read through our amendments and provide additional comments.
Comment: Title still says "study protocol"
Response: This has been amended, the title reads A Global Women's Rugby Union Web-Based Survey
Comment: Line 40: like in your abstract, you should initially state "rugby union" and then can abbreviate to rugby
Response: Thank you for highlighting this, line 39 where rugby is first stated in the introduction has been amended; Rugby Union (rugby) is a fast-paced collision sport...
Comment: Discussion is much improved
Response: Thank you for your previous comments to improve the quality of the manuscript.
Comment: Conclusion is acceptable
Comment: supplementary files were not viewable or hosted on the MDPI review page for separate download
Response: Apologies, the link to the supplementary files were included lines 317-319 which on receipt of this comment we have noticed are viewable in the word document but not when exported to PDF. Please see the links below. This has also been attached, please note my system keeps changing the attachment title to 'author-coverletter' which I am unable to change, this is a PDF of the supplementary files which has also been uploaded with the manuscript.
Player survey: Players survey
Coaches survey: Coaches survey
